# IL-33 via PKCμ/PRKD1 Mediated α-Catenin Phosphorylation Regulates Endothelial Cell-Barrier Integrity and Ischemia-Induced Vascular Leakage

**DOI:** 10.3390/cells12050703

**Published:** 2023-02-23

**Authors:** Deepti Sharma, Geetika Kaur, Shivantika Bisen, Anamika Sharma, Ahmed S. Ibrahim, Nikhlesh K. Singh

**Affiliations:** 1Integrative Biosciences Center, Wayne State University, Detroit, MI 48202, USA; 2Department of Ophthalmology, Visual and Anatomical Sciences, Wayne State University School of Medicine, Detroit, MI 48202, USA; 3Department of Biochemistry, Faculty of Pharmacy, Mansoura University, Mansoura 35516, Egypt; 4Department of Pharmacology, Wayne State University, Detroit, MI 48202, USA

**Keywords:** adherens junction, PKCμ/PRKD1, proliferative retinopathy, iBRB, vascular leakage, α-catenin

## Abstract

Angiogenesis, neovascularization, and vascular remodeling are highly dynamic processes, where endothelial cell–cell adhesion within the vessel wall controls a range of physiological processes, such as growth, integrity, and barrier function. The cadherin–catenin adhesion complex is a key contributor to inner blood–retinal barrier (iBRB) integrity and dynamic cell movements. However, the pre-eminent role of cadherins and their associated catenins in iBRB structure and function is not fully understood. Using a murine model of oxygen-induced retinopathy (OIR) and human retinal microvascular endothelial cells (HRMVECs), we try to understand the significance of IL-33 on retinal endothelial barrier disruption, leading to abnormal angiogenesis and enhanced vascular permeability. Using electric cell-substrate impedance sensing (ECIS) analysis and FITC-dextran permeability assay, we observed that IL-33 at a 20 ng/mL concentration induced endothelial-barrier disruption in HRMVECs. The adherens junction (AJs) proteins play a prominent role in the selective diffusion of molecules from the blood to the retina and in maintaining retinal homeostasis. Therefore, we looked for the involvement of adherens junction proteins in IL-33-mediated endothelial dysfunction. We observed that IL-33 induces α-catenin phosphorylation at serine/threonine (Ser/Thr) residues in HRMVECs. Furthermore, mass-spectroscopy (MS) analysis revealed that IL-33 induces the phosphorylation of α-catenin at Thr^654^ residue in HRMVECs. We also observed that PKCμ/PRKD1-p38 MAPK signaling regulates IL-33-induced α-catenin phosphorylation and retinal endothelial cell-barrier integrity. Our OIR studies revealed that genetic deletion of IL-33 resulted in reduced vascular leakage in the hypoxic retina. We also observed that the genetic deletion of IL-33 reduced OIR-induced PKCμ/PRKD1-p38 MAPK-α-catenin signaling in the hypoxic retina. Therefore, we conclude that IL-33-induced PKCμ/PRKD1-p38 MAPK-α-catenin signaling plays a significant role in endothelial permeability and iBRB integrity.

## 1. Introduction

Angiogenesis is a fundamental process that includes the generation of new blood vessels from pre-existing blood vessels. An appropriate balance between proangiogenic and anti-angiogenic factors highly controls this process. Nevertheless, abnormal angiogenesis or retinal neovascularization is associated with several ocular diseases leading to blindness. Retinal ischemia or hypoxia is a stimulus for neovascularization, leading to a compromised barrier function, dysfunctional and destabilized plexi, or hemorrhaging and retinal detachment [1,2,3,4]. Vascular endothelial growth factor (VEGF), a master regulator of permeability and angiogenesis, is primarily involved in neovascularization-associated ocular disorders. Hence, anti-VEGF therapeutics are presently employed in treating these diseases [5,6]. Although anti-VEGF therapies are effective in certain populations of patients, spontaneous or acquired resistance has been reported in a significant percentage of patients, indicating the association of other vasoactive mediators in the development of pathological angiogenesis in patients with ischemic retinopathies [7,8,9,10]. Several angiogenic cytokines, growth factors, and inflammatory mediators have been implicated in the progression of ocular diseases [11,12]. Therefore, considering the clinical issues associated with anti-VEGF therapies, investigating another effective mediator(s) might pave the way for new therapeutic options in neovascularization-related ocular diseases.

The blood-retinal barrier (BRB) is a highly dynamic and complex barrier that protects the retina from systemic immunological and inflammatory components while preserving retinal homeostasis. Due to the existence of junctional complexes (tight, adherens, and gap junctions) between endothelial and epithelial cells, BRB rigorously controls paracellular permeability [13]. As a result, changes in junction assembly and function significantly impact BRB characteristics, especially barrier permeability. The outer BRB (oBRB) and the inner BRB (iBRB) are the two separate barriers that make up the BRB. The iBRB dysfunction contributes to the pathophysiology of numerous retinal pathologies such as diabetic retinopathy (DR), retinal vein occlusion, retinopathy of prematurity (ROP), retinoblastoma, and retinitis pigmentosa [14,15,16]. The adherens junctions (AJs) of the iBRB include vascular endothelial (VE)-cadherin, p120, β-catenin, and α-catenin. The Ca^2+^-dependent transmembrane cell adhesion protein VE-cadherin has a conserved cytoplasmic tail that interacts with p120 and β-catenin. Furthermore, β-catenin binds to α-catenin and anchors the cadherin–catenin complex with actin [17]. In several cases, molecular mechanisms regulating the endothelial vascular permeability target the phosphorylation of AJs, their cleavage, and VE-cadherin internalization. Various agents, including histamine [18], tumor necrosis factor-α (TNF-α) [19], platelet-activating factor (PAF) [20], and VEGF [21], have been shown to affect AJ permeability and barrier function via phosphorylation of VE-cadherin and its binding partners at tyrosine residues. Several tyrosine kinases, including SRC kinase, c-SRC tyrosine kinase (CSK), and proline-rich tyrosine kinase 2 (PYK2), have been linked to VE-cadherin and β-catenin phosphorylation [22,23]. VEGF-induced SRC kinase activates the specific tyrosine (Y685) in the VE-cadherin cytoplasmic domain, thereby regulating angiogenesis and permeability [24,25]. In addition to VE-cadherin, VEGF also stimulated the tyrosine phosphorylation of other catenin proteins such as p120, and β-catenin, compromising cytoskeletal organization, cell–cell contact and impairing the barrier function of the endothelium [26,27].

Among various proinflammatory factors, IL-1 family members have been reported to increase oxidative stress, endothelial permeability, and BRB breakdown during the pathogenesis of retinal degenerative diseases [28]. IL-1β and IL-1α are the most widely explored for their angiogenic roles and contribution to retinal disease pathology [29]; however, the role of IL-33 in pathologies of ischemic retinopathies has not been much investigated. The angiogenic role of IL-33 is still under debate, as some reports show it has a proangiogenic role [30], while others exhibit its anti-angiogenic role [31]. In HUVECs, IL-33 stimulated endothelial NO production, resulting in increased endothelial permeability and loss of cadherin-mediated cell–cell contact, promoting angiogenesis [30]. Previously, we have shown that IL-33 regulates OIR/hypoxia-induced endothelial cell sprouting and retinal neovascularization [32]. Increased in vascular permeability loosens adherens junctions (AJs) to allow angiogenic sprouting [33]. Therefore, in the present study, we used human retinal microvascular endothelial cells (HRMVECs) and a murine model of oxygen-induced retinopathy (OIR) to understand the significance of IL-33-signaling on endothelial barrier disruption, leading to abnormal angiogenesis, and enhanced vascular permeability.

## 2. Materials and Methods

### 2.1. Reagents

Anti-α-catenin (sc-9988, dilution 1:500), anti-VE-cadherin (sc-9989, dilution 1:500), and anti-β-catenin (sc-7963, dilution 1:500) were obtained from Santa Cruz Biotechnology (Dallas, TX, USA). Anti-Phospho-p38 MAPK (4511), anti-Phospho-p44/42 MAPK (ERK1/2) (4370), anti-Phospho-SAPK/JNK (9255), anti-p38 MAPK (9212), anti-p44/42 MAPK (ERK1/2) (4695), anti-SAPK/JNK (9252), anti-Phospho-PKD/PKCμ (Ser916) (2051), anti-PKD/PKCμ (D4J1N) (90,039), anti-Phospho-PKD/PKCμ (Ser744/748) (2054), anti-Phospho-PKC (pan) (βII Ser660) (9371), anti-Phospho-PKCα/β II (Thr638/641) (9375), anti-Phospho-PKCδ (Thr505) (9374), anti-Phospho-PKCδ/θ (Ser643/676) (9376), anti-Phospho-PKCθ (Thr538) (9377), anti-Phospho-PKCζ/λ (Thr410/403) (9378), anti-PKCα (2056), anti-PKCζ (C24E6) (9368), anti-PKD/PKCμ (D4J1N) (90,039), and anti-PKCδ (D10E2) (9616) antibodies were obtained from Cell Signaling Technology (Beverly, MA, USA). VECTASHIELD Antifade mounting medium without DAPI (H-1700), Hoechst 33,342, Prolong Gold antifade reagent (P36984), and Alexa Fluor 568-conjugated goat anti-mouse immunoglobulin G were bought from Invitrogen (Carlsbad, CA, USA). Fluorescein isothiocyanate–dextran (average molecular weight 2,000,000) FD2000S-100MG (average molecular weight 70,000) and 46945-100MG-F, and Triton ™ X-100 (T9284) were obtained from Sigma-Aldrich (St. Louis, MO, USA). Normal Goat Serum Blocking Solution S-1000-20 was purchased from Vector Laboratories, Inc. (Burlingame, CA, USA).

### 2.2. Experimental Animals

Charles River Laboratories provided C57BL/6 mice (Wilmington, MA, USA). The Jackson Laboratory provided IL-33^flox/flox^ mice (#030619) and E2a-Cre mice (#003724). The mice were raised, housed, provided with ad libitum water and food in a 12-h light/12-h dark cycle setting. The animals were kept in the DLAR animal facility at Wayne State University Detroit, Michigan. For this investigation, male and female mouse pups aged postpartum day 12 (P12) to P17 were used. The Wayne State University Animal Care and Use Committee in Detroit, Michigan, approved each animal experiment.

### 2.3. IL-33 Knockout Mice Generation

We crossed IL-33^flox/flox^ mice [34] with E2a-Cre mice to obtain IL-33 knockout animals. In mice, the *Cre* recombinase triggers germ-line deletion of IL-33 [35]. The male and female mouse pups (IL-33^flox/flox^ and IL-33^−/−^) of age postpartum day 12 (P12) to P17 were used for the experiments.

### 2.4. Cell Culture

We purchased HRMVECs (ACBRI 181) from the Applied Cell Biology Research Institute. The cells were cultured in an EGM2 medium containing 0.25 μg/mL Amphotericin B and 10 μg/mL Gentamycin. The HRMVECs were maintained at 37 °C in an incubator with 95% air and 5% CO_2_.

### 2.5. Oxygen-Induced Retinopathy (OIR)

The 7-day old mice pups with dams were placed in a BioSpherix chamber and subjected to 75 ± 2% oxygen for 5 days (P7 to P12) before being returned to room air [36]. Control mice were littermates of the same age who were kept at ambient air (21% oxygen). The pups were euthanized at P13 and P15, their eyes were enucleated, retinas extracted, retinal tissue extracts were prepared, and analyzed by western blotting using the appropriate antibodies.

### 2.6. Measurement of OIR-Induced Vascular Leakage in Mice Retina

A novel high molecular weight fluorescein-dextran perfusion method was used for the measurement of vascular leakage in a mice model of oxygen-induced retinopathy (OIR) [36]. Briefly, mice pups and dams were placed in a BioSpherix chamber at P7 and subjected to 75% ± 2% oxygen for 5 days (P7 to P12) before being returned to room air [37]. Control mice were littermates of the same age who were kept at ambient air. At P17 mice pups were anesthetized and then perfused with fluorescein-conjugated high molecular weight dextran (2,000,000 molecular weight) through tail vein injection. The eyes were enucleated, fixed in 4% (*v*/*v*) paraformaldehyde (PFA) for 24 h at 4 °C. Then the retinas were isolated and a flat mount was prepared and visualized under a Zeiss LSM 800 confocal microscope (Carl Zeiss Microscopy, White Plains, NY, USA). The amount of leakage from iBRB is calculated as (IBRB − IB) × ABRB. Here, IBRB stands for average fluorescent intensity due to iBRB, IB for average fluorescent intensity from non-leaky area, and ABRB for area of iBRB leakage. Since there was little to no leakage in the peripheral retina, we used the peripheral retina as a non-leaky region for background removal. Data-normality tests were performed to exclude physiological differences in the animals’ distribution of FITC-dextran in the vasculature.

### 2.7. FITC-Dextran Flux Assay

To measure the endothelial permeability, HRMVECs were grown on the apical side of the Transwell insert. The cells were allowed to grow to form a monolayer. Cells were growth arrested overnight in serum-free media. Thereafter, FITC-conjugated dextran (~70,000 Da) at a working concentration of 100 μg/mL was added to the apical chamber. IL-33 (agonist) was added to both the apical and basal chamber for 2 h. In the case of inhibitor, cells were treated with inhibitor for 30 min before the agonist treatment. Then 100 μL of medium was transferred to 96-well plate from each (apical and basal) chamber and a BIOTEK SYNERGY H1 microplate reader (with Gen5™ Data Analysis Software v3.11, Santa Clara, CA, USA) was used to measure fluorescent intensity. The FITC-dextran flux was expressed as the % dextran diffused/h/cm^2^.

### 2.8. Electric Cell-Substrate Impedance Sensing (ESIC)

The effects of IL-33 on the real-time barrier function of human retinal endothelial cell monolayers were studied by measuring overall cellular impedance utilizing (ECIS^®^ Zθ (theta), Applied Biophysics Inc., Troy, NY, USA) technology. In brief, a 96-well array (96W20idf PET) was coated for 30 min with 100 µM cysteine (50 µL/well), followed by coating with 0.02% gelatin (50 µL/well) for 60 min. HRMVECs in EGM-2 were cultured and allowed to form a mature monolayer before being treated with several doses of IL-33 (1 ng/mL to 100 ng/mL). Following that, an alternating current of 1 μA was applied to an electrode at the bottom of the well to measure the total resistance (R) with regard to time and frequency. The optimum frequency corresponding to the maximum total R was chosen to be 4000 Hz based on our previous study [38]. The R value at each time point was adjusted to the baseline R before adding IL-33 and then displayed as a function of time. The data were collected throughout the experiment by calculating the area under the curve (AUC).

### 2.9. Cell-Surface Receptor Internalization

HRMVECs were allowed to rest in a serum-free medium overnight. Quiesced cells were treated with or without IL-33 (20 ng/mL) for the respective periods, then washed and incubated with sulfo-NHSS-SS-biotin in PBS for 30 min at 4 °C. After 30 min, 50 mM Tris (pH 8.0) was added to stop the reaction. The cells were lysed in lysis buffer, and the membrane proteins that had been biotinylated were affinity purified using avidin resins before being examined by western blotting. The receptor internalization is measured by the fraction of receptors present on the cell surface of control and agonist-treated cells.

### 2.10. Immunoprecipitation and Mass Spectroscopy

The quiesced retinal endothelial cells were treated for 60 min with IL-33 (20 ng/mL). Cells were washed before being lysed in lysis buffer and immunoprecipitated with anti α-catenin antibodies. The immunocomplexes were then incubated for 3 h at 4 °C with protein A/G beads. The bound proteins were eluted from the beads and separated on an SDS-PAGE gel before being stained with Coomassie Brilliant Blue R-250 to make the proteins visible. The α-catenin band was excised and digested in-gel with trypsin. At Wayne State University’s Protein/Molecular Structural Analysis Core, the resultant peptides were examined for the phosphorylated amino acid residues using LC-ESI-MS/MS.

### 2.11. Immunofluorescence

HRMVECs were grown to confluence in a six-well plate on a circular glass coverslip. The cells were quiesced overnight and then treated with IL-33 (20 ng/mL) for the indicated periods. In the case of inhibitor studies, cells were treated with the inhibitor for 30 min before the IL-33 treatment. The cells were fixed, permeabilized, blocked, and incubated with mouse anti-α-catenin and mouse anti-VE-cadherin (dilution 1:100) antibodies for overnight at 4 °C. The cells were then washed and incubated with Alexa Fluor 568-conjugated goat anti-mouse secondary antibodies (dilution 1:250) for 1 h at RT. The cells were counter-stained with DAPI (Hoechst 33,342, dilution 1:1000), washed, and mounted using VECTASHIELD mounting media. The images were captured using a Zeiss LSM 800 confocal microscope (Carl Zeiss Microscopy, White Plains, NY, USA).

### 2.12. Western Blotting

Electrophoresis was used to separate an equivalent quantity of protein from cell or tissue extracts on SDS-PAGE gels. The proteins that had been resolved on the gels were then electrophoretically transferred to a nitrocellulose membrane. Either 5% (*w*/*v*) nonfat dry milk or bovine serum albumin (BSA) was used to block the membrane. Appropriate primary antibodies were used to probe the nitrocellulose membranes. After washing, the membranes were incubated with horseradish peroxidase-conjugated secondary antibodies. The membrane antigen-antibody complexes were visualized using an improved chemiluminescent detection reagent (Supersignal West Pico Plus, Thermo Fisher Scientific, Waltham, MA, USA).

### 2.13. Statistics

The data were presented as the mean ± standard deviation (SD) of three independent experiments. To compare the differences between two groups, two-tailed t-tests were utilized. To investigate differences between more than two groups, one-way ANOVA with Tukey’s post hoc analysis was performed. We used GraphPad Prism 9 (Prism, Boston, MA, USA) for statistical analysis. All *p* values < 0.05 were significant.

## 3. Results

### 3.1. IL-33 Disrupts Human Retinal Endothelial Cell-Barrier Integrity

The role of IL-33 in HRMVECs barrier function in a real-time manner was investigated using electric cell-substrate impedance sensing (ECIS^®^, Applied Biophysics Inc., Troy, NY, USA) instrument. HRMVECs were treated with various concentrations of IL-33 (1, 10, 20, 50, and 100 ng/mL) when the (R) reached the plateau phase, where HRMVECs formed a stable and confluent monolayer with mature tight connections (Figure 1A). The barrier integrity of HRMVECs was then assessed based on total R over 10 h. As shown in Figure 1A, IL-33 treatment at 10, 20, 50, and 100 ng/mL resulted in impaired barrier functionality of HRMVECs with little or no effect at 1 ng/mL. The area under the curve (AUC) for each R curve was calculated to determine the influence of IL-33 on HRMVECs R throughout the experiment (Figure 1B). The dose-dependent impact of IL-33 revealed a substantial difference in AUCs compared to the control, lending credence to the hypothesis that IL-33 modulates cell R throughout the whole experiment. The effect of IL-33 on human microvascular endothelial cell-barrier function was further corroborated using fluorescein isothiocyanate (FITC)-labeled dextran flux assay. IL-33 enhanced HRMVECs barrier permeability relative to controls, as evaluated by a fluorescein isothiocyanate (FITC)-labeled dextran flow experiment (Figure 1C).

### 3.2. IL-33 Promotes α-Catenin Phosphorylation and Adherens Junction Disruption

Adherens junctions (AJs) are cell–cell adhesion complexes found in endothelial and epithelial cells that play critical roles in embryogenesis and tissue homeostasis [39,40,41]. To further understand how IL-33 increases EC barrier permeability, we examined the effect of IL-33 on endothelial adherens junction proteins. The steady-state levels of AJ proteins, particularly α-catenin, β-catenin, and VE-cadherin, were unaffected by IL-33 (Figure 2A,B). As a result, we postulated that IL-33 might disrupt AJs by inducing post-translational modifications of AJ proteins, triggering their separation from multimeric protein complexes. Following this viewpoint, we investigated the tyrosine (Tyr) and serine/threonine (Ser/Thr) phosphorylation of AJ proteins in HRMVECs. IL-33 increased Ser/Thr phosphorylation of α-catenin in HRMVECs in a time-dependent manner but had little or no effect on VE-cadherin or β-catenin phosphorylation (Figure 2B,C). The immunofluorescence staining of the HRMVECs monolayer treated with and without IL-33 for various time periods for α-catenin showed that it dissociates from the plasma membrane in response to IL-33 treatment (Figure 2C). α-catenins are essential cytoplasmic molecules that are hypothesized to connect the cadherin cytoplasmic domain to the actin cytoskeleton [42]. To further understand the effect of IL-33 on VE-cadherin endocytosis in HRMVECs, we performed a cell surface-receptor internalization experiment. We discovered that IL-33 promoted VE-cadherin endocytosis in HRMVECs (Figure 2D,E). Based on the data, it seems that IL-33 destroys endothelial AJs in HRMVECs via α-catenin phosphorylation. We next used mass spectrometry to demonstrate that IL-33 induces α-catenin phosphorylation at Thr654 residues in HRMVECs (Figure 3A–C).

### 3.3. PKCμ Regulates α-Catenin Phosphorylation and Endothelial Barrier Disruption

Protein kinase C (PKC) is a serine/threonine protein kinase that plays a range of functions in cell processes, including cell–cell adhesion. PKCα and PKCβ are shown to influence cell–cell junctions and permeability in vascular endothelial cells [43,44]. As a result, we sought to determine whether IL-33 activates any PKCs and, if so, its role in α-catenin phosphorylation and endothelial barrier disruption. The time-course experiment demonstrated that IL-33 increases PKCμ/PRKD1 phosphorylation at Ser^744/748^ residues (Figure 4A,B). We next investigated the effect of PKCμ/PRKD1 deficiency on α-catenin phosphorylation. PKCμ downregulation by its siRNA reduced IL-33-induced α-catenin phosphorylation, indicating that PKCμ modulates IL-33-induced α-catenin phosphorylation in HRMVECs (Figure 4C). The role of PKCμ in HRMEVCs barrier permeability and electrical resistance was then investigated. Figure 5A,B shows that siRNA-mediated PKCμ depletion effectively reduced IL-33-induced endothelial cell permeability. To confirm this, we found that PKCμ depletion in HRMVECs reversed not only IL-33-induced α-catenin dislocation from the plasma membrane (Figure 5C), but also abrogated IL-33-induced reduction in electrical resistance of HRMVECs (Figure 5D) evaluated by ECIS compared to the control group throughout the experiment, as indicated by the AUC in Figure 5E.

### 3.4. p38 MAPK Mediates IL-33-Induced Retinal Endothelial Cell-Barrier Disruption

Previous reports have suggested that MAPKs play a role in endothelial cell permeability in response to external cues [45,46]. Therefore, we investigated the role of MAPKs in IL-33-induced endothelial cell permeability. IL-33 enhanced the phosphorylation of JNK, ERK1/2, and p38 MAPK in a time-dependent manner in HRMVECs (Figure 6A). Based on these findings, we looked at the role of JNK, ERK1/2, and p38 MAPK in IL-33-induced HRMVECs barrier permeability. We used SP600125 to inhibit JNK activity, FR180204 to reduce ERK activity, and SB203580 to inhibit p38 MAPK activity. SP600125, a selective inhibitor of JNK [47], blocked IL-33-induced JNK phosphorylation in HRMVECs (Figure 6B). FR180204 is a competitive inhibitor of ERK, but it does not affect ERK phosphorylation by upstream kinases [48]. Similarly, SB203580 reduces p38 MAPK catalytic activity but does not affect p38 MAPK phosphorylation by upstream kinases [49]. Studies have shown that FR180204 and SB203580 inhibit AKT phosphorylation at serine 473 residue [48,50]. Therefore, we assessed the activity of FR180204 and SB203580 by looking at their effects on AKT phosphorylation in HRMVECs. Both FR180204 and SB203580 blocked IL-33-induced AKT phosphorylation in HRMVECs (Figure 6B). SB203580, a pharmacological inhibitor of p38 MAPK [49], attenuated IL-33-induced endothelial cell permeability (Figure 6C). We failed to observe any effect of SP600125 (JNK inhibitor) and FR180204 (ERK1/2 inhibitor) on IL-33-induced endothelial-cell permeability (Figure 6C). To confirm these results, we also investigated the role of SB203580 (p38 MAPK inactivation) on α-catenin localization at the plasma membrane. We observed that p38 MAPK inactivation inhibited IL-33-induced α-catenin phosphorylation and reversed the IL-33-induced dislocation of α-catenin from the plasma membrane (Figure 6D,E). We next tested the role of PKCμ in p38 MAPK activation. The siRNA-mediated downregulation of PKCμ blocked IL-33-induced p38 MAPK phosphorylation (Figure 6F), suggesting that p38 MAPK is downstream of PKCμ in IL-33-mediated retinal endothelial-cell permeability.

### 3.5. IL-33 Regulates iBRB Integrity in Oxygen-Induced Retinopathy (OIR)

We expanded our research on the function of IL-33 in OIR-induced retinal endothelial barrier breakdown to better comprehend the pathophysiological significance of our findings in HRMVECs. We genetically deleted IL-33 to examine its effect on OIR-induced retinal endothelial barrier dysfunction (Figure 7A,B). We observed an induced phosphorylation of PKCμ, p38 MAPK, and α-catenin in hypoxic retinas of IL-33^fl/fl^ mice, which was significantly inhibited in IL-33 knockout mice retina (Figure 7C,D). IL-33-deficient mice also showed a significant reduction in OIR-induced vascular leakage (FITC leakage) compared to IL-33^fl/fl^ mice (Figure 7E,F). These results suggest that IL-33 regulates hypoxia/ischemia-induced retinal endothelial barrier permeability, which is a predisposing cause for inner blood-retinal barrier (iBRB) damage and pathological retinal neovascularization.

## 4. Discussion

A critical component of various proliferative retinopathies is blood vessel dysfunction. The retina is susceptible to fluid buildup produced by excessive blood vessel leakage in the eye, and reducing this leakage is a treatment target in retinal disorders [51]. Here, we demonstrate that IL-33 induces retinal endothelial cell-barrier permeability in two independent models, HRMVECs and mouse oxygen-induced retinopathy (OIR) model. We have shown that IL-33 disrupts human retinal endothelial cell-barrier permeability through PKCμ-p38 MAPK-α-catenin signaling. We also observed that IL-33 via PKCμ-p38 MAPK-α-catenin signaling regulates OIR-induced vascular leakage in retinal vascular beds.

We and others have shown that IL-33, a stress-regulated cytokine produced largely by epithelial and endothelial cells, has a role in angiogenesis [32,52,53,54]. We have also shown that IL-33 has a role in post-ischemic neoangiogenesis [32]. Anti-VEGF medications used to treat pathological neovascularization (NV), but do not specifically block NV, as documented impairment in normal retinal vascular development and retinal function have also been noted [55]. During our studies, we observed that downregulation of IL-33 resulted in reduced NV without impairing intraretinal revascularization [32]. Therefore, depletion or blockage of IL-33 might be helpful in only regulating pathological retinal neovascularization without affecting regular retinal repair. Increased vascular permeability is frequently associated with the early stages of angiogenesis. Therefore, we looked for the role of IL-33 on retinal endothelial barrier permeability using electric cell–substrate impedance sensing (ECIS), and FITC-dextran flux assay. We observed that IL-33 treatment not only induces retinal endothelial cell permeability, but also decreases cell–cell adhesion in human retinal endothelial cells. We have previously shown that IL-33 regulates Jagged1-Notch1 mediated retinal endothelial cell sprouting and neovascularization [32]. The phosphorylation of adherens junction molecules, in particular VE-cadherin, increases the junctional permeability and sufficiently loosens the adhesions to permit efficient sprouting [33]. According to a computational model, the intercalation of cells that facilitates sprout elongation is made possible by regional variations in the adhesive characteristics between cells [56]. Therefore, we investigated how IL-33 affected the expression and phosphorylation of adherens junction proteins. Here, we observed that IL-33 induces Ser/Thr phosphorylation of α-catenin. From mass-spectrometric analysis, we found that IL-33 induces α-catenin phosphorylation at Thr-654 residue. Our immunofluorescence studies show that IL-33 treatment induces α-catenin dissociation from the plasma membrane, and this correlates with α-catenin phosphorylation and endothelial cell-barrier dysfunction. These findings suggest that IL-33-induced α-catenin phosphorylation at Thr-654 contributes to human retinal endothelial cell-barrier dysfunction. Various knockout studies have demonstrated that α-catenin is essential for tissue morphogenesis and cell–cell adhesion [57,58]. However, it is unknown how α-catenin controls cell-adhesion states or other dynamic adhesion modifications that result in angiogenesis.

There are several members of the PKC family, and each isozyme is strongly associated with a variety of cellular functions, including cell adhesion, cytoskeletal rearrangements, membrane traffic, and ion transport [59,60]. PKC regulates the endocytosis and internalization of several cell surface receptors [61,62,63]. In the present study, we observed that IL-33 induces PKCμ/PRKD1 activation in retinal endothelial cells, with little or no effect on other PKCs. According to previous studies, PKCα and PKCδ are crucial PKC isoforms in the Cholecystokinin (CCK)-dependent activation of PKCμ/PRKD1 in pancreatic acini [64]. Our findings suggest that PKCα and PKCδ might be involved in the phosphorylation of PKCμ/PRKD1 since we also observed induced phosphorylation of PKCα and PKCδ by IL-33 in HRMVECs. PKCμ/PRKD1 has been shown to regulate αvβ3 integrin trafficking and HUVECs migration [65]. The impact of PKCμ/PRKD1 on angiogenesis has also been reported [65]. However, the molecular pathways that PKCμ/PRKD1 activates in angiogenesis have yet to be fully discovered. The present study shows that IL-33-induced PKCμ/PRKD1 activation regulates α-catenin phosphorylation in HRMVECs. In addition, our observation also suggests a role for PKCμ/PRKD1 on IL-33-induced HRMEVCs barrier permeability and electrical impedance.

Mitogen-activated protein kinases (MAPKs) regulate numerous biological processes, including cell division, motility, differentiation, survival, and apoptosis [66,67,68]. As a result, the initiation and progression of various diseases depend on the dysregulation of MAPK signaling. The MAPK-signaling pathways are involved in the onset and development of cardiac and vascular diseases. It is activated by various extracellular stimuli, which leads to alterations in intracellular processes. Extracellular stimuli include cellular stress, adhesion molecules, neurohormones, and PKCs [69]. There is mounting evidence that many MAPK family members regulate signaling pathways, which ultimately results in several vascular disorders. The MAPK subfamilies-p38 MAPK, c-Jun NH2-terminal kinases (JNK 1, 2, and 3), and extracellular signal-regulated kinases (ERK1/2) regulate signaling in vascular diseases. The role of PKC-regulated MAPK signaling in diabetes complications has received a lot of attention. Several studies have shown that PKCs have a direct role in ERK and JNK activation, which results in cardiomyocyte hypertrophy and remodeling [70,71]. Under hypoxic conditions, the PKCμ/PRKD1-p38 MAPK signaling regulates melatonin-induced osteoblast development [72]. Furthermore, it has also been reported that PKCμ/PRKD1 mediated p38 MAPK phosphorylation regulates hypoxia-induced growth and metabolism of the SCC25 cells [73]. In this regard, our findings indicate that IL-33-induced PKCμ activation regulates p38 MAPK phosphorylation in HRMVECs. Furthermore, our results also emphasize that p38 MAPK activation regulates IL-33-induced retinal endothelial cell permeability.

Our recent study found that oxygen-induced retinopathy (OIR) induces IL-33 levels in hypoxic retinal endothelial cells. Additionally, OIR-induced retinal EC sprouting and neovascularization were prevented by the genetic deletion of IL-33 in mice. Several studies have shown that VEGFA not only stimulates the production of endothelial tip cells (sprouting) but also reduces adherens junctions (AJs) to promote vascular permeability [74]. It was also reported that tyrosine phosphorylation of VE-cadherin and c-SRC by VEGFR2 enhances junctional permeability and loosens adhesions enough to allow for efficient sprouting [33]. According to a computer model, local changes in adhesive affinities between cells allow the intercalation of cells that enable sprout extension [56]. Therefore, we investigated the involvement of PKCμ-p38 MAPK-α-catenin signaling in OIR-induced proliferative retinopathy and the effect of IL-33 deletion on this signaling. In the retina, OIR-induced PKCμ-p38 MAPK mediated α-catenin Ser/Thr phosphorylation, and IL-33 deficiency inhibited this signaling. We also observed that the genetic depletion of IL-33 results in reduced OIR-induced vascular leakage in the hypoxic retina. The findings here may provide more evidence for the significance of IL-33 in retinal endothelial cell permeability and neovascularization. Several knockout studies have shown that α-catenin plays a crucial role in tissue morphogenesis and cell–cell adhesion [57,58]. However, it is unknown how α-catenin regulates the dynamic cell adhesion states and the alterations that result in angiogenesis. Our findings may have highlighted the involvement of IL-33 and its signaling in hypoxia/ischemia-induced α-catenin phosphorylation and retinal endothelial cell-barrier failure.

## 5. Conclusions

In conclusion, our findings show that IL-33 controls hypoxia/ischemia-induced retinal endothelial permeability via the PKCμ-p38 MAPK-α-catenin pathway. These findings suggest that increased IL-33 levels in the hypoxic retina promote endothelial barrier rupture and vascular leakage, contributing to the pathogenesis of angiogenesis-dependent proliferative retinopathies.

## Figures and Tables

**Figure 1 cells-12-00703-f001:**
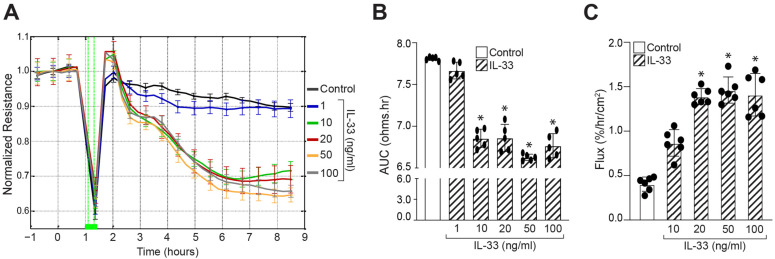
IL-33 regulates human retinal endothelial cell-barrier integrity. (**A**) HRMVECs normalized resistance over time in the presence or absence of IL-33 was measured at a frequency of 4000 Hz. At t = 0 h, IL-33 treatments were applied, and resistance was measured on the ECIS electrode until the indicated time periods. (**B**) The area under the normalized resistance curve over the range of t = 0 h to t = 9 h for each group was statistically analyzed. (**C**) The quiescent HRMVECs monolayer was treated with control (vehicle) or with indicated concentrations of IL-33, and dextran flux was measured. Statistical analysis was performed using the ANOVA test followed by Tukey’s post hoc test. * *p* ≤ 0.05.

**Figure 2 cells-12-00703-f002:**
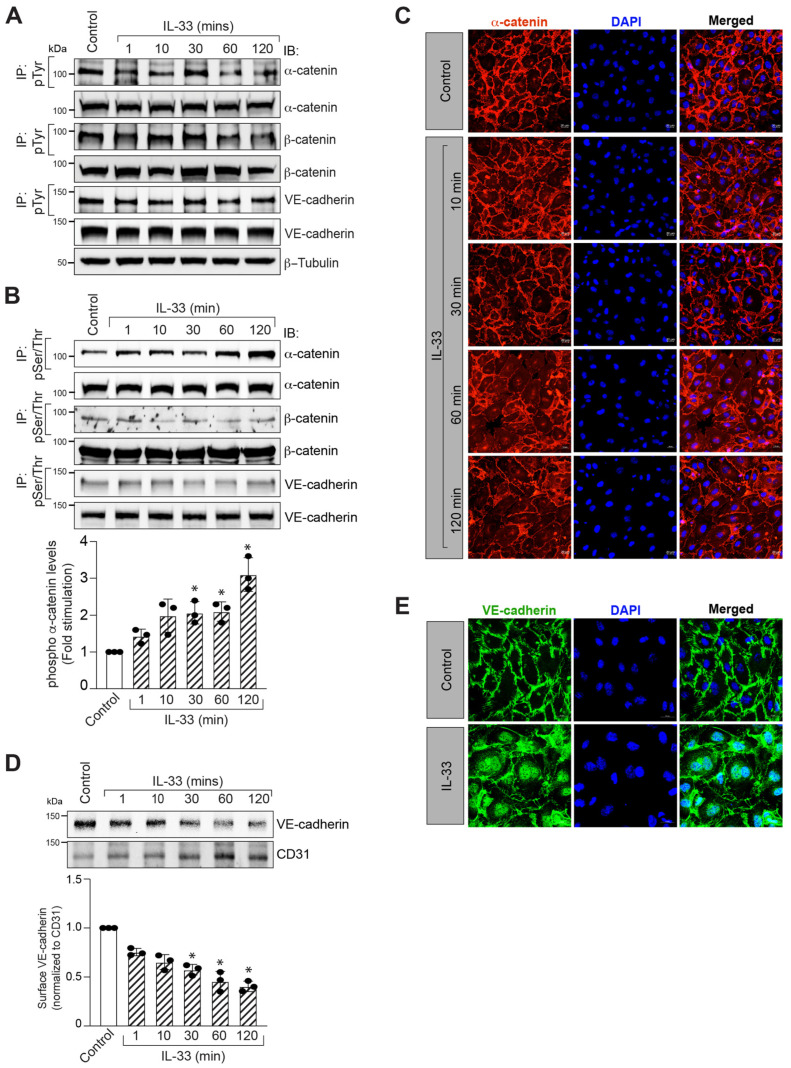
IL-33 disrupts retinal endothelial cell adherens junctions (AJs) by Ser/Thr phosphorylation of α-catenin. (**A**,**B**) Quiescent HRMVECs monolayers were treated with control or IL-33 (20 ng/mL) for the indicated periods and cell extracts were prepared. An equal amount of cell extracts was immunoprecipitated (IP) with anti-phosphotyrosine (pTyr) or anti-phospho serine/threonine (pSer/Thr) antibodies, and the immunocomplexes were analyzed by western blotting (WB) for the indicated AJ proteins. We also examined the cell extracts in panels A and B for the indicated AJ protein levels followed by normalization to β-tubulin. (**C**) HRMVECs monolayers were treated with or without IL-33 for the indicated time periods, fixed, and probed with mouse anti-α-catenin antibodies followed by Alexa Fluor 568-conjugated goat anti-mouse secondary antibodies. Fluorescence images were acquired with a Zeiss LSM800 confocal microscope. (**D**) Quiescent HRMVECs were treated with or without IL-33 (20 ng/mL) for the indicated time periods, cell-surface proteins were biotinylated, cell-surface fractions were separated using avidin resin and analyzed by WB for cell surface VE-cadherin levels. (**E**) Everything is the same as in panel C, except the cells were treated with and without IL-33 for 2 h and probed with mouse anti-VE-cadherin antibodies. The bar graphs show quantitative analysis of three independent experiments, expressed as mean ± SD. *, *p* < 0.05 versus vehicle control. Scale bar, 20 μm.

**Figure 3 cells-12-00703-f003:**
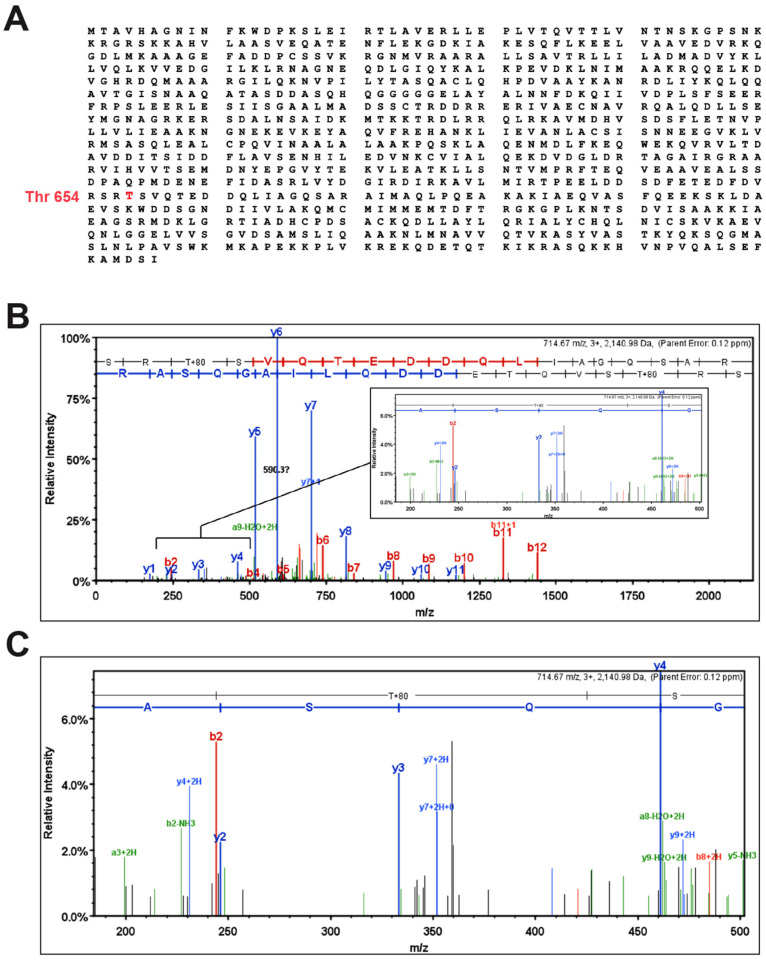
IL-33 induces the phosphorylation of α-catenin at Threonine 654 residue. (**A**) The amino acid sequence of the α-catenin protein is given, coupled with a phosphorylated threonine amino acid residue (Thr654) verified by tandem mass spectrometry. (**B**,**C**) Quiescent HRMVECs were treated for 60 min with IL-33 (20 ng/mL), and cell extracts were prepared. Cell extracts were immunoprecipitated with anti-α-catenin antibodies, resolved on SDS-PAGE, stained with Coomassie Brilliant Blue, and the α-catenin molecular mass band was subjected to high-resolution mass-spectrometric analysis. A full MS/MS scan with a zoomed-in region is displayed for a clearer view of the assigned b and y ions for the 3+ charge state parent ion at 714.67 m/z.

**Figure 4 cells-12-00703-f004:**
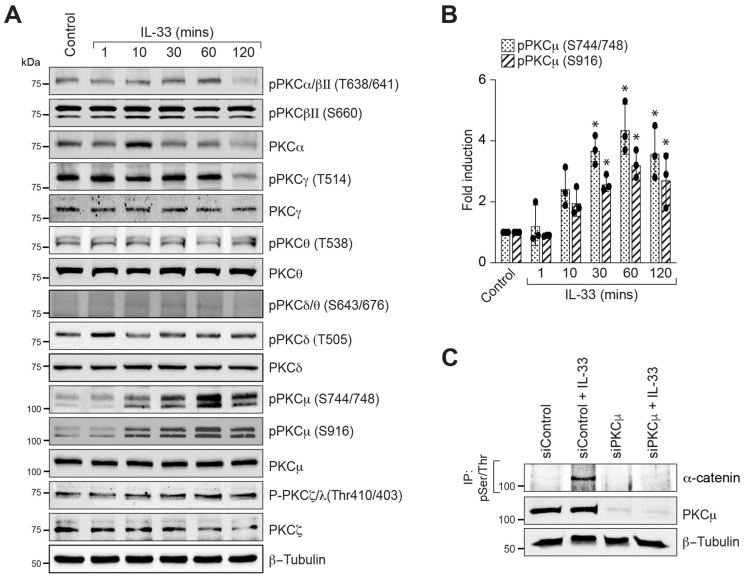
PKCμ regulates IL-33-induced α-catenin phosphorylation. (**A**) Quiescent HRMVECs were treated with or without IL-33 (20 ng/mL) for the respective periods, and phospho-PKC and total PKC levels were determined. (**B**) The bar graphs depict the quantitative analysis of three separate experiments given as mean ± SD. (**C**) HRMVECs were transfected with either control siRNA (siControl) or PKCμ siRNA (siPKCμ), quiesced, and treated with IL-33 for 60 min. The cell extracts were prepared and immunoprecipitated (IP) using anti-phospho serine/threonine (pSer/Thr) antibodies, and the immunocomplexes were tested for α-catenin antibodies. PKCμ levels in cell extracts were also determined using specific antibodies and normalized to β-tubulin. * *p* < 0.05 versus control.

**Figure 5 cells-12-00703-f005:**
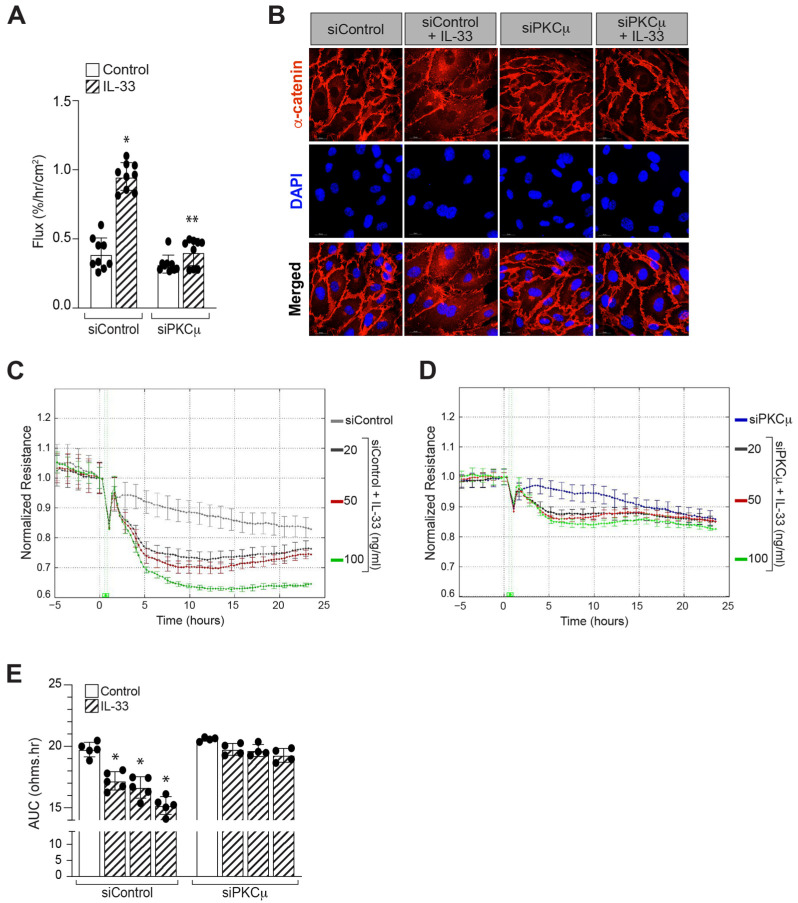
PKCμ mediates IL-33-induced endothelial AJ disruption. (**A**) Quiescent HRMVECs monolayers were treated with or without of IL-33 (20 ng/mL), and FITC-dextran flux was measured. (**B**) HRMVECs were transfected with control siRNA (siControl) or PKCμ siRNA (siPKCμ), quiesced, treated with or without IL-33 for the indicated time periods, fixed, and probed with mouse anti-α-catenin antibodies followed by Alexa Fluor 568-conjugated goat anti-mouse secondary antibodies. Fluorescence images were captured using an LSM800 Zeiss confocal microscope. (**C**,**D**) HRMVECs were transfected with siControl or siPKCμ, quiesced, and treatments were applied at t = 0, and resistance were measured on the ECIS electrode. (**E**) The area under the normalized resistance curve over the range of t = 0 h to t = 25 h for each group was statistically analyzed. * *p* ≤ 0.05 versus siControl, ** *p* ≤ 0.01 versus siControl + IL-33. Scale bar represents 20 μm.

**Figure 6 cells-12-00703-f006:**
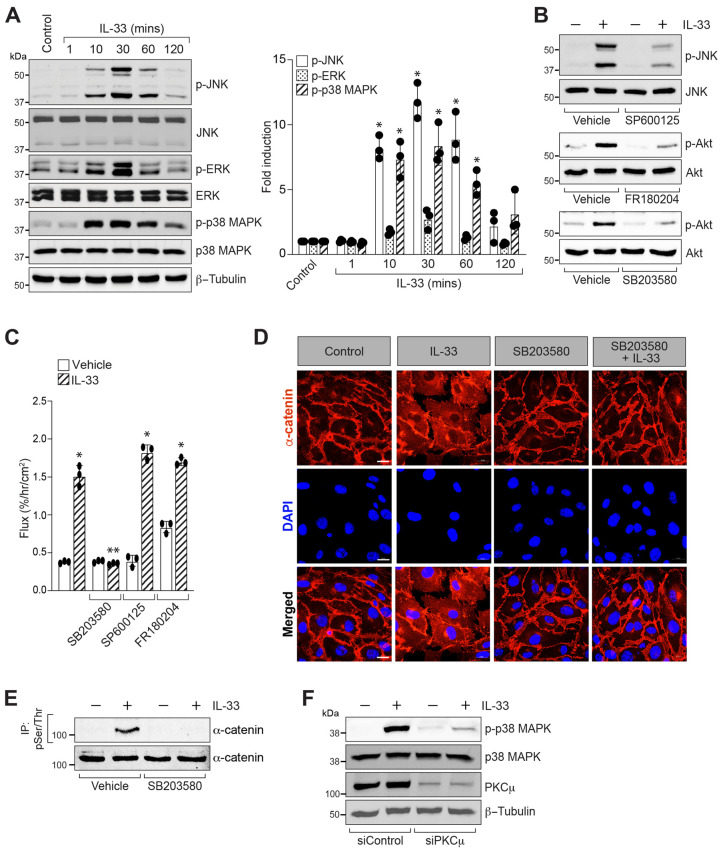
p38 MAPK mediates IL-33-induced endothelial AJ disruption. (**A**) Quiescent HRMVECs were treated with or without IL-33 (20 ng/mL) for the indicated periods before being analyzed for phospho JNK (p-JNK), phospho ERK1/2 (p-ERK1/2), and phospho p38 MAPK (p-p38 MAPK) using their specific antibodies. The blots were reprobed for the total levels of JNK, ERK1/2, p38 MAPK, and normalized to β-tubulin. The bar graphs show quantitative analysis of three independent experiments, expressed as mean ± SD. (**B**) Quiescent HRMVECs were incubated with JNK inhibitor (SP600125), ERK inhibitor (FR180204), and p38 MAPK inhibitor (SB203580) for 30 min and then treated with or without IL-33 (20 ng/mL) for 30 min. The cell extracts were analyzed by western blotting for the indicated proteins. (**C**) The quiescent HRMVECs monolayer was incubated with p38 MAPK inhibitor (SB203580) for 30 min and then treated with or without IL-33 (20 ng/mL) for 2 h, and dextran flux was measured. (**D**) Everything is the same as in panel B, except that the HRMVECs were treated with SB203580, fixed, and probed with mouse anti-α-catenin antibodies followed by Alexa Fluor 568-conjugated goat anti-mouse secondary antibodies. Fluorescence images were acquired with a Zeiss LSM800 confocal microscope. (**E**) Quiescent HRMVECs were incubated with SB203580 for 30 min and then treated with or without IL-33 (20 ng/mL) for 2 h. The cell extracts were analyzed by western blotting for the indicated proteins. The cell extracts were prepared and immunoprecipitated (IP) using anti-phospho serine/threonine (pSer/Thr) antibodies, and the immunocomplexes were tested for α-catenin antibodies. α-catenin levels in cell extracts were also determined using specific antibodies. (**F**) HRMVECs were transfected with control siRNA (siControl) or PKCμ siRNA (siPKCμ), quiesced, and treated for 30 min with or without IL-33 (20 ng/mL). Following the treatment period, cell extracts were prepared, and western blotted for phospho p38 MAPK levels. The blot was reprobed for p38 MAPK levels and normalized to β-tubulin. * *p* ≤ 0.05 versus Control, ** *p* ≤ 0.01 versus Control + IL-33.

**Figure 7 cells-12-00703-f007:**
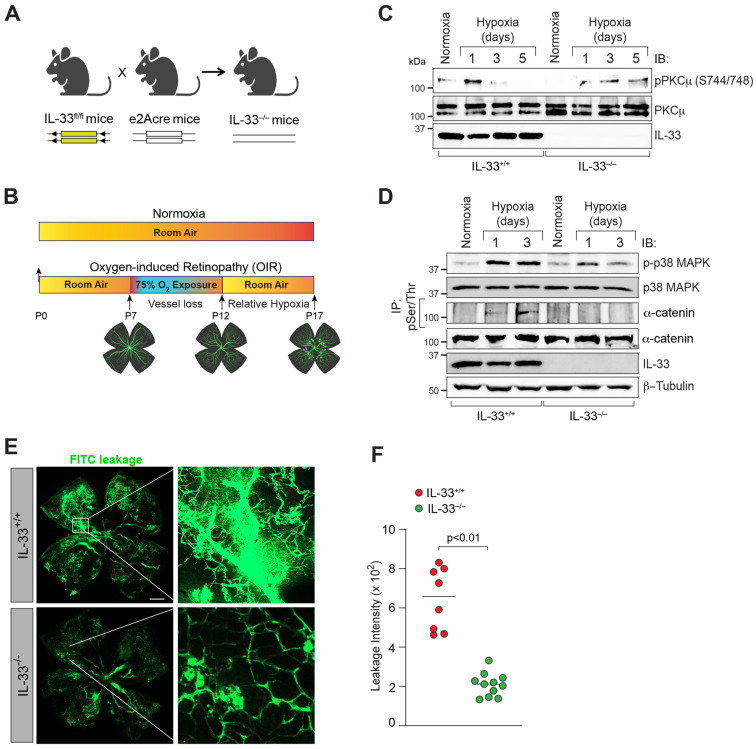
IL-33-induced PKCμ-p38 MAPK-α-catenin signaling regulates OIR-induced vascular leakage in hypoxic retina. (**A**) The breeding approach for producing IL-33 knockout mice is depicted schematically. (**B**) A sketch of the mouse OIR model. C57BL/6 mice pups with mothers were exposed to 75% oxygen from P7 to P12 before being returned to the room. (**C**,**D**) Eyes were enucleated at P13, P15, and P17, retinas were extracted, retinal tissue extracts were prepared, and analyzed by western blotting for the indicated proteins using their specific antibodies and normalized to β-tubulin. The retinal tissue extracts were also immunoprecipitated (IP) using anti-phospho serine/threonine (pSer/Thr) antibodies, and the immunocomplexes were tested for α-catenin antibodies. *n* = 6 mice per group. (**E**) At P17, mice pups were anesthetized, perfused with FITC-dextran, sacrificed, eyes enucleated, retinas were isolated, and whole retinal mounts were examined for vascular leakage. (**F**) The dot blot depicts a quantitative examination of vascular leakage as mean ± SD. Each dot symbolizes a mouse’s eye. Scale bar represents 500 μm.

## Data Availability

Data can be made available via contacting corresponding author.

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
