# Peer review of "IL-33 via PKCμ/PRKD1 Mediated α-Catenin Phosphorylation Regulates Endothelial Cell-Barrier Integrity and Ischemia-Induced Vascular Leakage"

_cells, 2023, doi:10.3390/cells12050703_

Round 1
Reviewer 1 Report
Manuscript title: PKCμ/PKD1 mediated α-Catenin phosphorylation regulates endothelial cell barrier integrity and Ischemia-induced vascular leakage
In this manuscript, authors used human retinal microvascular endothelial cells (HRMVECs) and a murine model of oxygen-induced retinopathy (OIR) to study the function of IL-33 on retinal endothelial barrier disruption. IL-33 destroys endothelial AJs by phosphorylation of α-catenin at Thr654 residue in HRMVECs. The authors also observed that genetic deletion of IL-33 resulted in reduced vascular leakage in the hypoxic retina. Both the in vitro and in vivo data show IL-33-induced PKCμ-p38 MAPK-α-catenin signaling plays a significant role in endothelial permeability and iBRB integrity. This study is very important for better understanding the role of IL-33 in endothelial permeability. However, I have a few comments that need further attention prior to resubmission.
1. In the title, I searched on NCBI, PKCμ is also known as PRKD1 which locates on chromosome 14. PKD1, which locates on chromosome 16, seems no relation to your study. I also did not see it or its explanation in other places except together with PKCμ. Is it a spell mistake or does it have other meanings?
2. From reading your paper, IL-33 seems the main character in your research. I am curious why it is not included in your title. If possible, I suggested adding IL-33.
3. In your method part 2.6 line 147, you used dextran with 2,000,000 molecular weight to analyze the permeability in mice. I thought 2,000,000 is too high to leak out of the endothelial barrier even in high permeability situation (PMID: 29071198). Could you please list the references you read, in which they used this dextran to complete this experiment?
4. From line264-265, you mentioned that IL-33 promoted VE-cadherin endocytosis in HRMVECs. Could you please add the IF results showing the expression of VE-cadherin reduced on the plasma and increased in the cytoplasm, as the WB results show no change of total VE-cadherin. Thanks.
5. In Figure 2C, we could see the expression of α-catenin reduced upon IL-33 stimulation in IF results. But from the WB in Figure 2A, total α-catenin consistent. How to explain this?
6. Figure 3A legends, please indicate the meaning of high-lighted aa sequences.
7. Figure 5B, we could see the line of the scale, but not the value. Please modify it.
8. Could you please add the WB results of the phosphorylation of α-catenin at Thr654 after p38 MAPK inhibitor treatment with or without IL-33 induction?
9. Have you checked the angiogenesis in IL-33 knockout mice? In line 398-399, you mentioned IL-33 has a role in angiogenesis, so we are expecting defect of vascular system and more permeability in IL-33 knockout mice. How to explain the results of less permeability you showed when IL-33 is knocked out? You may would like to add it in discussion. Thanks.
10. IL-33 is an extra-cellular cell factor. PKCμ is a serine/threonine kinase mainly localize in cytoplasm. What is the receptor between IL-33 and PKCμ?
11. Please discuss the advantages and shortcomings of targeting IL-33 compared to targeting IL-1, VEGF or others.
Author Response
Reviewer #1
- In the title, I searched on NCBI, PKCμ is also known as PRKD1 which locates on chromosome 14. PKD1, which locates on chromosome 16, seems no relation to your study. I also did not see it or its explanation in other places except together with PKCμ. Is it a spell mistake or does it have other meanings?
Answer: We are sorry for the confusion. PKCμ is also known as PRKD1 or Protein Kinase D1 (PKD1). PKCμ/PRKD1/PKD1 is located on chromosome 14. The reviewer is confusing Protein Kinase D1 (PKD1) with Polycystic Kidney Disease 1 (PKD1). They both are abbreviated as PKD1. The Polycystic Kidney Disease 1 (PKD1) is located on chromosome 16. To avoid the confusion, we now changed PKD1 to PRKD1 in the revised manuscript.
- From reading your paper, IL-33 seems the main character in your research. I am curious why it is not included in your title. If possible, I suggested adding IL-33.
Answer: In response to Reviewer #1’s suggestions, we have now modified the title in the revised manuscript.
- In your method part 2.6 line 147, you used dextran with 2,000,000 molecular weight to analyze the permeability in mice. I thought 2,000,000 is too high to leak out of the endothelial barrier even in high permeability situation (PMID: 29071198). Could you please list the references you read, in which they used this dextran to complete this experiment?
Answer: We are thankful to the reviewer for asking this question. Studies have shown that low-molecular-weight FITC-dextran (40000-50000) is not suitable for retina, as it leaks out of the vasculature to stain the entire retina in whole-mount preparations (Microvasc Res.1993; 46(2):135-142). The authors have also shown that mouse retinal vessel abnormality can be assessed using high-molecular-weight FITC-dextran (2,000,000). In our lab itself we have used various molecular weight FITC-dextrans and concluded that high molecular weight FITC-dextran (2,000,000) is best for assessing retinal vasculature, vascular integrity, and leakiness. Therefore, our group as well as others have used high-molecular-weight FITC-dextran (2×106) to observe OIR-regulated retinal vessel abnormality (Exp Eye Res, 2007; 84: 529-536; Mol. Vis. 2011; 17: 3566-3573; Blood, 2010; 116:1377-1385; J Biol Chem. 2011; 286: 22489-22498).
- From line264-265, you mentioned that IL-33 promoted VE-cadherin endocytosis in HRMVECs. Could you please add the IF results showing the expression of VE-cadherin reduced on the plasma and increased in the cytoplasm, as the WB results show no change of total VE-cadherin. Thanks
Answer: In response to Reviewer #1’s suggestions, it is now included as Fig. 5E in the revised manuscript.
- In Figure 2C, we could see the expression of α-catenin reduced upon IL-33 stimulation in IF results. But from the WB in Figure 2A, total α-catenin consistent. How to explain this?
Answer: We are thankful to the reviewer for pointing this out to us. We have not observed any reduced expression of α-catenin by IL-33 in our experiments. We have replaced the IF images in figure 2A to address Reviewer’s #1 concern.
- Figure 3A legends, please indicate the meaning of high-lighted aa sequences.
Answer: In response to Reviewer #1’s suggestions, we have removed the high-lighted aa sequences in Figure 3A.
- Figure 5B, we could see the line of the scale, but not the value. Please modify it.
Answer: In response to Reviewer #1’s suggestions, we added it in the revised manuscript.
- Could you please add the WB results of the phosphorylation of α-catenin at Thr654 after p38 MAPK inhibitor treatment with or without IL-33 induction?
Answer: We have now included it in the revised manuscript. Kindly refer to Fig. 6E.
- Have you checked the angiogenesis in IL-33 knockout mice? In line 398-399, you mentioned IL-33 has a role in angiogenesis, so we are expecting defect of vascular system and more permeability in IL-33 knockout mice. How to explain the results of less permeability you showed when IL-33 is knocked out? You may would like to add it in discussion. Thanks.
Answer: Angiogenesis is a highly coordinated process involving multiple phases, including endothelial cell (EC) activation, vascular basement membrane degradation, increase vascular permeability, vascular sprouting, EC migration, proliferation, and EC tube formation. Vascular permeability is the initial step of angiogenesis, where the adherens and tight junctions between the ECs are broken so that ECs can migrate, proliferate, and form vessels (Angiogenesis 2008; 11: 109-119). In a previous paper from my lab, we have shown that IL-33 regulates OIR/hypoxia-induced endothelial cell sprouting and retinal neovascularization (Commun Biol. 2022; 5: 479). In the present manuscript, we have shown that IL-33 increases vascular permeability, which is consistent with our previous observations that IL-33 induces angiogenesis. Therefore, knockdown of IL-33 reduces both permeability and angiogenesis in murine OIR models. We have now discussed it in the revised manuscript (Kindly refer to Page 16, lines 577-583).
- IL-33 is an extra-cellular cell factor. PKCμ is a serine/threonine kinase mainly localize in cytoplasm. What is the receptor between IL-33 and PKCμ?
Answer: We have previously shown that IL-33 regulates intracellular signaling via its receptor ST2 (Commun Biol. 2022; 5: 479). Therefore, it may be suggested that IL-33 via ST2 receptor regulate PKCm phosphorylation.
- Please discuss the advantages and shortcomings of targeting IL-33 compared to targeting IL-1, VEGF or others.
Answer: It is now discussed in the revised manuscript (Kindly refer to Page 15, lines 469-475).
Reviewer 2 Report
Very well performed study and well written manuscript! Can be accepted in the current form.
Author Response
Thank you
Reviewer 3 Report
Summary: Authors Sharma et al., are trying the investigate the significance of IL-33 on retinal endothelial barrier disruption using a murine model of oxygen-induced retinopathy (OIR) and human retinal microvascular endothelial cells (HRMVECs). They concluded that genetic deletion of IL-33 reduced the vascular leakage and OIR-induced PKCmu-p38 MAPK-alpha-catenin signaling in the hypoxic retina.
Comments:
- The authors provided sufficient introduction.
- The rationale of the study: Can the authors highlight the reasons for them in study the significance of IL-33 in particular in endothelial permeability and iBRB integrity. There are many other signaling molecules that control the endothelial permeability and angiogenesis. Why IL-33 in particular? Can the authors elaborate on the novelty of the study.
- Do the authors know of any physiologically active antagonists that are available to suppress IL-33.
- Since, the main emphasis of this article is IL-33, I advise that the authors should consider altering the title to include IL-33.
- The authors made a conclusion that, the genetic deletion of IL-33 reduced vascular leakage. But there is a published literature by Augustine et al (PMID:31796062)., which says that “IL-33 deficiency causes persistent inflammation and severe neurodegeneration in retinal detachment. Doesn’t this contradict the current articles conclusion? Can the authors justify these differences?
- I would suggest the authors to bold the sub figures (eg: A, B etc should be bolded) in Fig 2 in consistent with the rest of the figures in the manuscript.
- I would suggest the authors to include a graphical flow chart of their hypothesis and the findings (conclusion) as fig.8 for easier understanding to the readers.
- What is the age of the IL-33flox/flox mice and E2a-Cre mice used in this study. I would suggest that the authors should consider including these details in the experimental animal’s methods section.
- There are several published reports which state that, in the mouse OIR model, nursing mothers were less tolerant to hypoxic conditions compared to the pups. Did the authors observe any such differences in this study?
Author Response
Reviewer #3
- The authors provided sufficient introduction.
Answer: Thank you.
- The rationale of the study: Can the authors highlight the reasons for them in study the significance of IL-33 in particular in endothelial permeability and iBRB integrity. There are many other signaling molecules that control the endothelial permeability and angiogenesis. Why IL-33 in particular? Can the authors elaborate on the novelty of the study.
Answer: Previously, we have shown that IL-33 regulates OIR/hypoxia-induced endothelial cell sprouting and retinal neovascularization (Commun Biol. 2022; 5: 479). Vascular permeability leads to EC migration and angiogenic sprouting. Therefore, we looked for the role of IL-33 in the endothelial permeability and iBRB integrity.
- Do the authors know of any physiologically active antagonists that are available to suppress IL-33.
Answer: Soluble IL-1 receptor-like-1 (sST2) is a decoy receptor for IL-33 and studies have shown that local overexpression of sST2 inhibits IL-33 signaling (Am J Respir Cell Mol Biol. 2013; 49: 552-562). In addition, it was also shown that sST2 overexpression induces IL-10 expression. Studies have demonstrated that IL-10 promotes retinal angiogenesis (PLoS One. 2008; 3: e3381). Therefore, in our studies we have not used sST2 to inhibit IL-33-induced signaling.
- Since, the main emphasis of this article is IL-33, I advise that the authors should consider altering the title to include IL-33.
Answer: We have included it in the revised manuscript.
- The authors made a conclusion that, the genetic deletion of IL-33 reduced vascular leakage. But there is a published literature by Augustine et al (PMID:31796062)., which says that “IL-33 deficiency causes persistent inflammation and severe neurodegeneration in retinal detachment. Doesn’t this contradict the current articles conclusion? Can the authors justify these differences?
Answer: We are thankful to reviewer #2 for pointing out this. Augustine et al. (PMID:31796062) have shown that genetic depletion of IL-33 exacerbated sodium hyaluronate-induced inflammatory responses. In addition, they also demonstrated that Müller cells from IL-33-/- mice express lower levels of CCL2 and IL-6 compared to WT mice, particularly under hypoxic conditions. In this manuscript, we have demonstrated that IL-33 regulates OIR/hypoxia-induced vascular leakage and neovascularization in a mouse model of relative hypoxia. Therefore, our observations align with Augustine et al.'s findings in hypoxic conditions.
- I would suggest the authors to bold the sub figures (eg: A, B etc should be bolded) in Fig 2 in consistent with the rest of the figures in the manuscript.
Answer: We have corrected it in the revised manuscript.
- I would suggest the authors to include a graphical flow chart of their hypothesis and the findings (conclusion) as fig.8 for easier understanding to the readers.
Answer: We included a graphical abstract with our revised manuscript.
- What is the age of the IL-33flox/flox mice and E2a-Cre mice used in this study. I would suggest that the authors should consider including these details in the experimental animal’s methods section.
Answer: We included it in the revised manuscript (Kindly refer to Page 3, lines 132-133).
- There are several published reports which state that, in the mouse OIR model, nursing mothers were less tolerant to hypoxic conditions compared to the pups. Did the authors observe any such differences in this study?
Answer: We have not experienced this with our mouse OIR model. Still, I was aware of it and asked the same question to the BioSpehrix (the hyperoxia/hypoxia chamber providers) technician while purchasing the chamber. He informed me that we must regularly calibrate the oxygen sensor to avoid these things. Because of continuous use, the oxygen sensor calibration becomes inaccurate, displaying 75% O2 flow to the chamber, whereas the actual O2 flow to the chamber is substantially higher. When nursing mothers from uncalibrated chambers are exposed to room air, they develop severe hypoxia and, in some cases, die too.
Reviewer 4 Report
The manuscript by Sharma et al explores the role of IL-33 in regulating barrier function in retinal MVEC and following hypoxic injury in the retina. Using a mix of in vitro and in vivo studies, the authors show that treatment of RMVEC with IL-33 leads to decreased barrier function, increased VE-cadherin internalization, and ser/thr phosphorylation of a-catenin. They go on to show that this phosphorylation is likely due to activation of PKCµ/p38 using siRNA and inhibitors. In vivo, genetic depletion of IL-33 reduces the number of leaky vessels after hypoxic damage, and that the activation of PKCµ and phosphorylation of a-catenin also are reduced by loss of IL-33. Overall, the manuscript lays out a logical set of experiments which support a role for an IL-33/PKCµ/p38 pathway in oxygen-induced retinopathy and ocular vascular disease. However, there are some concerns that should be addressed before publication.
1. The authors use iv injection of FITC -dextran to measure in vivo leak. The methods section indicates that the quantification of leak was done using a equation IBRB-IB x ABRB. The authors should define how they selected a “non-leaky area” for background subtraction. In addition, they should indicate whether this analysis accounts for differences in the distribution and loading of the dye (i.e. the dye concentration in the vessels) between animals.
2. A control for the inhibitors used in Fig 6B is missing. What is the relative specificity and efficiency of each inhibitor for its target kinase in your system?
3. Figure 4A examines the activation of various PKC isoforms by IL-33. Clearly, PKCµ phosphorylation at S744/748 is upregulated. However, phosphorylation of S916 is known to correlate with activation, yet the phosphorylation of this residue decreases, then returns to baseline upon increasing doses of IL-33. The authors should address this discrepancy.
4. Relatedly, the author state in the discussion (line 426) that IL-33 had little to no effect on other PKCs, which is not an accurate statement based on data shown in fig 4A. Potential roles for other PKC isoforms should be discussed.
5. In Figure 7, Western blot data shows that 1 day of hypoxia stimulates PKCµ phosphorylation in wildtype animals, but that 3 days of hypoxia also stimulates phosphorylation in IL-33 knockouts, suggesting that this PKC-mediated response might only be delayed in these animals, not absent. Critically, as leak data (Fig 7D) was gathered at 5 days, this timepoint must also be included in the blots in fig 7C in order to support the author’s conclusions.
6. The authors previously published that IL-33 deficiency causes reduced sprouting and neoangiogenesis in the retina. As this could also affect the amount of leak (at least the ABRB), the potential contribution of reduced sprouting to the barrier function data should be discussed.
7. a-catenin Thr-654 has been previously recognized as a phosphorylation site for casein kinase (Escobar et al J Cell Sci 2015). In the absence of in vitro phosphorylation assays or rescue with phospho-mimetic a-catenin, the authors should discuss the strength of their evidence that PKCµ directly phosphorylates this site (consensus seq? could PKCµ be upstream of CK1/2?).
8. The authors should include a discussion of what is known about the signaling between PKCµ and p38.
9. Line 414, Thr-654 is not phosphorylated by IL-33.
10. Line 418, “is responsible” is not supported by the data, change to “contributes to”.
11. Lines 428 and 450 are missing references.
Author Response
Reviewer #4
- The authors use iv injection of FITC -dextran to measure in vivo leak. The methods section indicates that the quantification of leak was done using a equation IBRB-IB x ABRB. The authors should define how they selected a “non-leaky area” for background subtraction. In addition, they should indicate whether this analysis accounts for differences in the distribution and loading of the dye (i.e. the dye concentration in the vessels) between animals.
Answer: We used the peripheral retina as a non-leaky area for background subtraction, as we observed little or no leakage in the peripheral retina. We observed minor differences in the distribution of FITC within the vessels due to individual physiological variations between the animals. We have considered these variations while analyzing the data. We performed normality tests and eliminated outliers from the study (Kindly refer to Page 4, lines 165-168).
- A control for the inhibitors used in Fig 6B is missing. What is the relative specificity and efficiency of each inhibitor for its target kinase in your system?
Answer: We checked the effect of the inhibitors during our studies but did not include that data in the manuscript. The data on inhibitors is now in the revised manuscript. SP600125, a selective inhibitor of JNK (PNAS, 2001; 98:13681-13686), blocked IL-33-induced JNK phosphorylation in HRMVECs. FR180204 is a competitive inhibitor of ERK, but it does not affect ERK phosphorylation by upstream kinases (J Biol Chem. 2015; 290: 20972-20983). Similarly, SB203580 reduces p38 MAPK catalytic activity but does not affect p38 MAPK phosphorylation by upstream kinases (Biochem Biophys Res Commun. 1999; 263(3): 825-831). Studies have shown that FR180204 and SB203580 inhibit AKT phosphorylation at serine 473 residue (J Biol Chem. 2015; 290: 20972-20983 & J Biol Chem. 2000; 275:7395-7402). Therefore, we assessed the activity of FR180204 and SB203580 by looking at their effects on AKT phosphorylation in HRMVECs. Both FR180204 and SB203580 blocked IL-33-induced AKT phosphorylation in HRMVECs (Kindly refer to Page 10, lines 373-374; Page 11, lines 377-384).
- Figure 4A examines the activation of various PKC isoforms by IL-33. Clearly, PKCµ phosphorylation at S744/748 is upregulated. However, phosphorylation of S916 is known to correlate with activation, yet the phosphorylation of this residue decreases, then returns to baseline upon increasing doses of IL-33. The authors should address this discrepancy.
Answer: We thank reviewer #3 for bringing this to our attention. We analyzed our data and observed that PKCµ phosphorylation at Ser916 correlates with Ser744/748 phosphorylation. We also performed new experiments to confirm these findings. We have included this data in the revised manuscript (Kindly refer to Fig. 4A).
- Relatedly, the author state in the discussion (line 426) that IL-33 had little to no effect on other PKCs, which is not an accurate statement based on data shown in fig 4A. Potential roles for other PKC isoforms should be discussed.
Answer: We have now included it in the revised manuscript (Kindly refer to Page 15, lines 504-508).
- In Figure 7, Western blot data shows that 1 day of hypoxia stimulates PKCµ phosphorylation in wildtype animals, but that 3 days of hypoxia also stimulates phosphorylation in IL-33 knockouts, suggesting that this PKC-mediated response might only be delayed in these animals, not absent. Critically, as leak data (Fig 7D) was gathered at 5 days, this timepoint must also be included in the blots in fig 7C to support the author’s conclusions.
Answer: In response to Reviewer #3’s suggestions, we have now included 5-day hypoxia data in the revised manuscript (Kindly refer to Fig. 7C).
- The authors previously published that IL-33 deficiency causes reduced sprouting and neoangiogenesis in the retina. As this could also affect the amount of leak (at least the ABRB), the potential contribution of reduced sprouting to the barrier function data should be discussed.
Answer: In response to Reviewer #3’s suggestions, we have included it in the revised manuscript (Kindly refer to Page 16, lines 577-583).
- a-catenin Thr-654 has been previously recognized as a phosphorylation site for casein kinase (Escobar et al J Cell Sci 2015). In the absence of in vitro phosphorylation assays or rescue with phospho-mimetic a-catenin, the authors should discuss the strength of their evidence that PKCµ directly phosphorylates this site (consensus seq? could PKCµ be upstream of CK1/2?).
Answer: We were aware of this study by Escobar et al. In this study, Escobar et al. demonstrated that casein Kinase 1 or 2 (CK1/CK2) modulates a-catenin phosphorylation at S652/S641 residue. They hypothesized that CK1/CK2 modulates a-catenin phosphorylation at T654, although they were not sure about it. Furthermore, no evidence has been provided in the article that a-catenin phosphorylation at T654 modulates intercellular adhesion.
- The authors should include a discussion of what is known about the signaling between PKCµ and p38.
Answer: In response to Reviewer #3’s suggestions, we have included it in the revised manuscript (Kindly refer to Page 16, lines 568-571).
- Line 414, Thr-654 is not phosphorylated by IL-33.
Answer: We corrected it in the revised manuscript.
- Line 418, “is responsible” is not supported by the data, change to “contributes to”.
Answer: We corrected it in the revised manuscript.
- Lines 428 and 450 are missing references.
Answer: We have provided references in the revised manuscript.
Round 2
Reviewer 1 Report
The authors have addressed all my concerns. Thanks.
Reviewer 4 Report
Revision is acceptable.